# A Comprehensive Overview of Stress, Resilience, and Neuroplasticity Mechanisms

**DOI:** 10.3390/ijms26073028

**Published:** 2025-03-26

**Authors:** Mario Humberto Buenrostro-Jáuregui, Sinuhé Muñóz-Sánchez, Jorge Rojas-Hernández, Adriana Ixel Alonso-Orozco, German Vega-Flores, Alejandro Tapia-de-Jesús, Perla Leal-Galicia

**Affiliations:** 1Departamento de Psicología, Universidad Iberoamericana Ciudad de México, Ciudad de México 01219, Mexico; smsanch@gmail.com (S.M.-S.); biol.jorge.rojas@gmail.com (J.R.-H.); ixelao@yahoo.com.mx (A.I.A.-O.); perla.leal@correo.uia.mx (P.L.-G.); 2Facultad de Psicología, Universidad Nacional Autónoma de México, Ciudad de México 04510, Mexico; 3Ciencias de la Salud, Universidad Internacional de Valencia, 46002 Valencia, Spain; german.vega.f@professor.universidadviu.com; 4Educación, Universidad Internacional de La Rioja, 26006 Logroño, Spain; 5Departamento de Salud, Universidad Iberoamericana Ciudad de México, Ciudad de México 01219, Mexico; alejandro.tapia@ibero.mx

**Keywords:** stress resilience, neuroplasticity, early-life stress, allostatic load, adaptation

## Abstract

Stress is a core concept in the mental health field that expands upon the seminal definition of stress as an acute response to the disruption of homeostasis. Stress is a complex process that involves both environmental challenges and the triggering of internal responses and impacts physiological, psychological, and behavioral systems. The capacity of the human brain to cope with stress is particularly crucial in early life, when neurodevelopment is highly plastic. Early-life stress (ELS), defined as exposure to severe chronic stress during sensitive periods of development, has been shown to cause lasting changes in brain structure and function. However, not all individuals exposed to ELS develop pathological outcomes, suggesting the presence of resilience mechanisms: adaptive processes that allow an individual to cope with adverse situations while maintaining psychological and neurobiological health. The aim of this review was to synthesize recent advances in the understanding of the neuroplasticity mechanisms underlying resilience to ELS. We discussed the neurobiological pathways implicated in stress response and adaptation, including the roles of neurogenesis, synaptic plasticity, and neural circuit remodeling. By focusing on the interplay between stress-induced neuroplastic changes and resilience mechanisms, we aimed to provide insights into potential therapeutic targets for stress-related psychopathology.

## 1. Introduction

To date, the study of adverse experiences in neuroscience has focused primarily on stress, exploring the effects of various stressors on behavior and neuronal alterations. However, recent advancements in psychology have shown that some individuals exposed to extremely adverse situations, such as physical, psychological, or sexual abuse, develop mechanisms that enable them to maintain their mental health. These mechanisms have been referred to using various terms, including “resilient mechanisms”. However, there is no consensus on the definition of the term “resilience”. Some definitions describe a process that enables individuals to overcome negative events [1,2]. This discussion of definitions is explored in greater depth later in this article.

Stress, a process in which an event or series of events—both internal and external—that are highly challenging, uncontrollable, and overwhelming activates the mechanisms necessary to restore equilibrium or homeostasis, is closely related to resilience. The physiological stress response is adaptive in acute situations and critical during life-threatening events [3]. In response to environmental demands, biological systems adjust to maintain physiological processes within normal ranges—a process known as allostasis, which is defined as active regulation to maintain these processes [4]. However, excessive, prolonged, or chronic allostasis can become ineffective, leading to a harmful allostatic load that increases stress vulnerability and the risk of developing disease [5].

Stress encompasses a series of processes involving the perception and evaluation of challenging stimuli and the associated responses. Figure 1 shows a general outline of the main components and processes involved in the stress and resilience phenomena. This conceptualization divides the phenomenon into the following five main components [3]:External stimuli out of the subject’s control that impose demands on the organism;Neural processes that evaluate these demands and available resources;Physiological, behavioral, and subjective activations indicative of stress;Neuroadaptations in brain systems involved in emotion and motivation under chronic stress;Cognitive, physiological, and behavioral adaptations in response to stressors.

**Figure 1 ijms-26-03028-f001:**
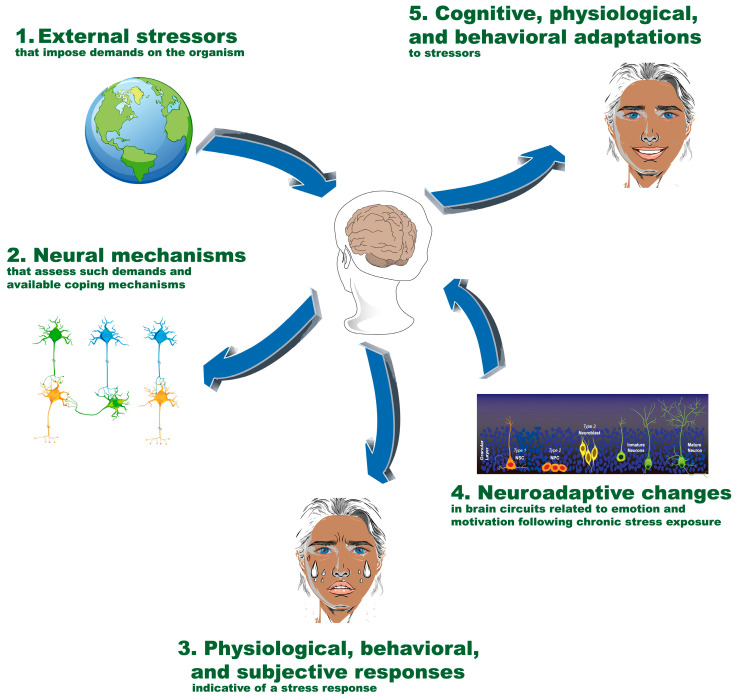
Main components and processes involved in stress and resilience. The figure illustrates the five main components involved in the stress and resilience process: (**1**) external stimuli that impose demands on the organism; (**2**) neural processes that assess these demands and available resources; (**3**) physiological, behavioral, and subjective stress-related responses; (**4**) neuroadaptations in brain systems related to emotion and motivation under chronic stress; and (**5**) cognitive, physiological, and behavioral adaptations to stressors. These components interact dynamically to shape an individual’s stress response and resilience. NSC: neural stem cells; NPC: neural progenitor cells.

External events or stimuli that place demands or strain on the organism refer to situations or events that occur outside the individual and are uncontrollable, such as interpersonal conflicts, the loss of a relationship, or the death of a loved one. On the other hand, internal stressors originate within the individual and may include factors like hunger, sleep deprivation, and extreme temperatures. Neural processes evaluate the environmental demands and the availability of adaptive resources to cope with them (evaluation). The perception and evaluation of stress depend on specific aspects of external or internal stimuli, personality traits, the availability of internal resources, the prior emotional state, and specific brain regions, such as the amygdala, hippocampus, insula, and orbitofrontal, medial prefrontal, and cingulate cortices, which are involved in the perception and evaluation of emotional and stressful stimuli [3,6]. Physiological, behavioral, and subjective activations signal stress to the organism. Physiological responses manifest through two main stress pathways: the activation of the hypothalamic–pituitary–adrenal axis (HPA), which releases corticotropin-releasing factor (CRF) from the paraventricular nucleus (PVN) of the hypothalamus in the presence of stressor stimuli and activates the anterior pituitary gland to liberate adrenocorticotrophic hormone (ACTH), which will travel throughout the bloodstream to the suprarenal gland that releases cortisol, and the autonomic nervous system. CRF exerts an extensive influence on extrahypothalamic regions and plays a critical role in modulating subjective and behavioral responses to stress [3,7]. Neuroadaptations in brain systems involved in emotion and motivation under chronic stress can lead to long-term changes in brain systems involved in emotion and motivation, which are reviewed further below. Cognitive, physiological, and behavioral adaptations in response to stressors refer to how the organism attempts to cope with stress to restore homeostasis. These processes can involve behavioral changes like implementing coping strategies, cognitive modifications to re-evaluate the stressful situation, and physiological adjustments to counteract the effects of stress on the body [1,3,8]. The effectiveness of these adaptations influences the overall impact of stress on an individual’s health and well-being.

Environmental stressors can have lasting effects depending on the life stage in which they occur. Adverse experiences in early childhood can result in long-term negative consequences; however, under certain circumstances, the brain can generate plasticity that mitigates negative effects even later in life [9]. Chronic exposure to elevated stress levels is associated with increased susceptibility to mood disorders, anxiety, and addictions. Conversely, some studies suggest that moderate stress levels during early developmental stages can foster a sense of control, improving the perception of mastery and promoting resilience [10,11].

The focus of this review is to understand how resilient systems overcome stress in general and early stress in particular. For these purposes, elucidating the interactions among neurochemical, genetic, and epigenetic processes over time, as well as their relationships with adaptive mechanisms—especially those associated with neuroplasticity—that facilitate successful resilient responses is essential. In this review, we analyzed the concepts of early stress and resilience, the models employed to study these phenomena, and the coping mechanisms associated with resilience. Additionally, we examined the brain structures and systems implicated in motivated behavior and the related neuroplasticity mechanisms. We also explored how this growing body of knowledge can contribute to the development of a stress-resilient profile. Finally, we discussed how identifying the neurobiological components of resilience could improve approaches to the prevention and treatment of stress-related disorders.

## 2. Stress

Stress, arousal, and anxiety are interrelated but distinct psychological constructs that impact cognition, behavior, and performance. Briefly, stress is a process involving the perception of and response to challenging or threatening stimuli [12,13]. Arousal is an acute physiological component of the stress-related response, activating the autonomic nervous system and releasing hormones to prepare the body for action [14]. In contrast, anxiety can arise as a consequence of prolonged or repeated stress and is linked to specific emotional states and brain processes [14,15]. Chronic stress patterns induce anxiety-like behaviors. In summary, arousal is an immediate physiological response to stress, whereas anxiety can be an emotional state that follows chronic stress. In particular, the definition of stress has evolved over time in association with advancements in the field. Selye (1936) first identified stress as a nonspecific response observed in rats during his experiments, which he termed “general adaptation syndrome”, encompassing the organism’s generalized response to a threat [16]. This concept has since been expanded to include the ability to adapt to the surrounding environment and to anticipate situations that may pose challenges to survival or safety [17]. Importantly, the term “stress” has been used interchangeably in the literature to refer to both the physiological response to a threatening stimulus and the stimulus itself. In this work, stress refers to the physiological response of an organism to a stimulus that potentially endangers its integrity, as well as the repercussions of that response on brain function and behavior, particularly when a threatening situation occurs during early developmental stages.

Exposure to stress during early life (early-life stress [ELS]) triggers a cascade of neuroendocrine responses, primarily through the activation of the HPA axis. Stress exposure can be acute or chronic. Sustained glucocorticoid release, which is characteristic of chronic stress, has detrimental effects on brain regions involved in emotional regulation, including the prefrontal cortex (PFC), ventral hippocampus, and amygdala [9,18]. Maternal separation from neonates for specific durations during the early postnatal period is widely employed as an ELS model, particularly in rodents. Studies of animal models have shown that maternal separation of rats from postnatal day 1 to day 14 induces chronically increased cortisol levels that persist into adulthood [19,20,21]. This phenomenon has also been observed in other animals, such as rhesus monkeys [22], and even in humans [23]. Early-life adverse experiences can shape the size and function of several brain areas. In children living in institutions and developing under deprivation, lower volumes in brain regions such as the medial prefrontal, inferior frontal, and inferior temporal areas are observed [24]. These children also have lower intelligence quotients and a high prevalence of attention-deficit/hyperactivity disorder, even after being adopted by nurturing families [24]. Although the volume of brain regions involved in complex processes such as memory or executive functions decreases in children experiencing deprivation in institutionalized environments, the volume of the amygdala increases, which can lead to high emotional susceptibility to stressful experiences later in life [25]. This evidence indicates the impact of adverse experiences early in life on brain function.

Stress is present throughout life; however, its impact may vary depending on the intensity and the age of exposure. Early life is characterized by heightened synaptic plasticity, rendering the developing brain particularly vulnerable to the detrimental effects of excessive stress. Critical developmental periods, during which environmental stimuli shape neural circuits, can be disrupted by chronic stress, leading to maladaptive outcomes. For instance, prolonged exposure to glucocorticoids during childhood has been shown to dysregulate the HPA axis, resulting in either hyperreactivity or hypoactivity in adulthood [26,27]. This topic is explored in greater detail later in this article. Figure 2 presents a general scheme of the activation of the hypothalamic–pituitary–adrenal (HPA) axis in response to a stress-triggering event.

### 2.1. Early-Life Stress

Stress can manifest early in life under various circumstances, significantly impacting brain development. During early developmental stages, the HPA axis is influenced by external factors, particularly the maternal environment. The HPA axis matures progressively and undergoes a transient hyporesponsive period, occurring between postnatal days 4 and 14 in rodents. This period is characterized by low but stable circulating corticosterone (CORT) levels and reduced sensitivity to stressors [28]. Early life stages are critical determinants of the eventual phenotype of an individual, playing a key role in programming physiological responses that are activated in later developmental stages [29]. During this phase, an organism perceives its environment and adapts its phenotype accordingly, a phenomenon known as early-life programming [30]. This early-life programming will shape the future stress response.

Several authors have proposed and tested hypotheses to explain the outcomes of ELS and how these outcomes might determine whether an individual becomes vulnerable or resilient to stress in adulthood [31,32,33]. Two primary hypotheses addressing this topic are the match/mismatch hypothesis [34] and the cumulative stress hypothesis [35], which are summarized below.

#### 2.1.1. Match/Mismatch Hypothesis

The match/mismatch hypothesis suggests that adverse early-life experiences trigger adaptive processes, enabling an individual to better cope with future aversive challenges [33]. According to this hypothesis a “match” occurs when the stressful environment experienced in adulthood aligns with the type of stress encountered in early life. In such cases, individuals are expected to exhibit enhanced adaptation and resilience because their physiological and behavioral systems have been “programmed” to handle such stressors. Conversely, a “mismatch” arises when a discrepancy exists between the early and adult environments. This mismatch can lead to increased vulnerability to psychopathology, as the early adaptations may be ineffective or even detrimental in a different context [31,35]. Several studies, mainly in animal models, provide evidence for the match/mismatch hypothesis. For instance, animal studies have shown that mild early-life stress can improve coping abilities in the face of similar adult stressors. Female mice exposed to mild early-life stress displayed better coping with chronic psychological stress in adulthood. Rats experiencing neonatal bacterial infection were less affected by inescapable tail shock stress in adulthood, showing a blunted corticosterone response and fewer depression-like behaviors [33]. Rats with low levels of maternal licking and grooming showed improved hippocampus-dependent learning and long-term potentiation (LTP) under high-stress conditions in adulthood [36]. Similarly, maternal deprivation led to enhanced hippocampal synaptic plasticity and emotional learning under high adult stress [37]. However, those findings were not always consistent, suggesting that more ways to respond to stress exist. In this regard, the cumulative stress hypothesis has been proposed and suggests that the accumulation of adverse experiences increases vulnerability, which also has important support that is described below.

#### 2.1.2. Cumulative Stress Hypothesis

In contrast, the cumulative stress hypothesis is a deterministic psychopathological model and proposes that deviations from social norms define what is considered pathological [32]. According to this hypothesis, the accumulation of adverse experiences early in life predisposes individuals to greater vulnerability to future aversive challenges, including physical and mental disorders. The cumulative effects of stress exposure across the lifespan increase the allostatic load, meaning that the greater the stress exposure is, the worse the outcome [35]. This heightened allostatic load increases the likelihood of developing pathologies [33]. This hypothesis suggests a dose-dependent relationship between stress exposure and risk of disease; the more stressors an individual encounters, the higher their vulnerability becomes. Population studies have shown that a greater cumulative number of stressful life events is significantly predictive of alcohol and drug use disorders, even after accounting for various control factors [38]. This result suggests that the more adversity an individual experiences, the greater their risk of developing addiction [3]. Also, studies using the chronic mild stress (CMS) paradigm in rats demonstrate that prolonged exposure to a variety of mild stressors can lead to anhedonia, a core symptom of depression. While some animals show resilience, many develop this condition, indicating that the accumulation of even mild stressors can have significant behavioral effects [39]. An interesting question related to the cumulative stress hypothesis is to evaluate whether the accumulation or the intensity of the stimulus modulates the brain adaptations and the behavioral response, and if the combination of both can be employed as a predictor of susceptibility for mental health disorders.

#### 2.1.3. Three-Hit Concept

Building on these frameworks, Daskalakis et al. (2013) developed the three-hit concept, which integrates genetic predisposition (Hit 1), the early-life environment (Hit 2), and the later-life environment (Hit 3) [31]. This theory aims to integrate the role of heredity with the effects of environmental stimuli, suggesting that ELS modulates the genetic programming of specific brain circuits underlying the emotional and cognitive aspects of behavioral adaptation to subsequent stressors. This model highlights the complex interplay between genetic and environmental factors, offering a comprehensive perspective on the long-term impacts of ELS. Figure 3 illustrates our integration of the three hypotheses that have attempted to explain the outcomes of ELS and the possible psychological, physiological, and behavioral responses of individuals in adulthood.

### 2.2. Early Stress and Critical Developmental Periods

In this context, the interaction between environmental stimuli and the brain can shape diverse neural circuits, especially when it occurs during critical developmental periods from the perinatal period through adolescence, and can determine the increase in the capacity for adaptation or drive to vulnerability in response to adverse conditions later in life. Critical periods are sensitive windows in development where an environmental stimulus has profound and lasting effects on the brain and behavior [40]. Adversity is a fundamental concept for understanding the topics we will explore later. It refers to difficult and harmful events or conditions that threaten an individual’s well-being [31,33]. Adversity includes a wide range of experiences, ranging from the difficulty and unpredictability of the environment to psychosocial stressors such as abuse, neglect, loss, and conflict [41]. Exposure to adversity, particularly in early life, can have lasting effects on development and health, increasing susceptibility to mental and physical health problems [31,41,42]. Adversity during these key moments significantly impacts neurodevelopment and the risk of mental disorders, whereas outside these windows, its impact is lower. Gillespie et al. (2009) reported that childhood abuse alters the development of the HPA axis and its connections with the amygdala, predisposing individuals to depression in adulthood. In rodents, exposure to glucocorticoids during sensitive periods affects the fear response. These effects are modulated by the quality of maternal care [43]. Furthermore, some cases of recovery suggest that experience-dependent plasticity mechanisms, such as glial cell-mediated myelination, shape brain development even outside sensitive periods [40]. Here, we briefly summarize the main critical periods of development related to stress. We suggest that readers with a deeper interest in the subject consult the study by Ho and King [40].

During the perinatal stage, the intrauterine environment is the primary source of stimuli. Factors such as maternal stress and adversity affect fetal brain development possibly through plasticity mechanisms such as neurogenesis and synaptogenesis [40,44,45,46]. Perinatal stress in rats drives the disruption of the selective permeability of blood–brain barrier [47], and increases the number of ramified microglia and their brain distribution in adulthood [48]. These results suggest that stress in this stage can modify the brain balance and immune response, leading to susceptibility to alterations in brain homeostasis. In rodents, prenatal adversity has been shown to generate anhedonic behaviors in offspring [49]. In humans, prenatal stress increases the risk of mental disorders, including depression [44,50], and alters connectivity between the amygdala and the PFC [51]. Moreover, increased maternal cortisol levels are related to increased amygdalar connectivity in newborn girls [52]. Prenatal stress also affects epigenetics. In rodents, a decrease in histone acetylation has been observed, leading to decreased gene transcription in the hippocampus [53], whereas in humans, prenatal stress has been linked to the methylation of genes associated with glucocorticoids [54]. Maternal adversity prior to conception can influence the brain development of offspring, as evidenced by the reduced gray matter volume in children of mothers with a history of child abuse [55].

During the first year of life, experience-dependent plasticity is crucial. Caregivers are the main source of not only stimuli but also adversity, with infants under one year old being the most vulnerable to maltreatment, according to reports from the U.S. Department of Health and Human Services [56]. Care deprivation, such as physical or emotional neglect, a lack of food, or low cognitive stimulation, at this stage can decrease the number of synaptic connections [57]. For example, institutionalized children show disorganized attachment styles, reduced brain volume, and functional differences detected by electroencephalography (EEG) [24,58]. Child maltreatment affects the amygdala–hippocampus–prefrontal circuit, which is related to fear learning and emotional regulation [59]. In a study of 262 children aged between 8 to 16 years who were exposed for two years to maltreatment, such as physical, emotional, and sexual abuse, as well as domestic violence, the authors reported that these children exhibited precocious puberty and signs of premature cellular aging, such as deoxyribonucleic acid (DNA) methylation, which were correlated with the prevalence of depressive symptoms [60]. In animal models, a lack of maternal interaction alters fear learning and amygdala activation, whereas maternal presence blocks these stress responses [61].

Stress induced in children living in war situations during childhood is a complex scenario due to the severity of the stressful experiences. The circumstances accompanying armed conflict violate the safety and integrity of children, exposing them to all types of violence to themselves and their loved ones, as well as forced displacement, which involves leaving their place of origin, friends, belongings, etc. All these circumstances are highly threatening from a biological point of view and can induce a wide variety of stress reactions. A study evaluating minors who migrated to southern Germany, either accompanied or alone, revealed the presence of stress signs such as nightmares and physiological and psychological reactivity, including irritability/anger and sleep disorders [62]. In a study analyzing post-traumatic stress symptoms in children aged 6 to 18 years living in or near war-torn countries such as Burundi, the Democratic Republic of the Congo, Iraq, Palestine, Tanzania, and Uganda, a self-report questionnaire was administered, and the presenting symptoms differed between children (6 to 12 years) and adolescents (13 to 18 years): the younger group exhibited more avoidance and dissociative symptoms, whereas adolescents presented more hypervigilance and intrusive thoughts [63]. These extreme stress situations can also favor the development of resilience mechanisms, which is addressed later in this work.

Childhood is a critical period of brain development, and even less menacing situations can act as stressors that can shape neural circuitry. In childhood, entering school introduces new sources of stimuli, such as teachers and peers, which can be both enriching and adverse [64]. Children in unpredictable environments exhibit lower effortful control, greater cognitive flexibility, and biases toward negative stimuli [65]. Also, in a study of Sprague Dawley rats, researchers found that chronic exposure to threats, such as bullying, impacts brain development, leading to increased dendritic spine growth in the amygdala and atrophy in the hippocampus and PFC [66].

Adolescence is characterized by a second wave of synaptogenesis and synaptic pruning, with strong hormonal and social influences. The attachment system facilitates emotional regulation, but interactions with peers become more relevant, promoting social adaptations and stress responses [67]. The stress recalibration hypothesis posits that a positive environment during puberty can mitigate early adversities [68]. However, the persistence of adverse environments can alter the HPA axis [69].

Based on the evidence presented here, we can conclude that stress, especially during early developmental stages, has significant and lasting impacts on the brain and behavior. During these critical periods of high synaptic plasticity, the brain is particularly vulnerable to adverse stimuli, which can alter fundamental processes such as neurogenesis, synaptogenesis, and HPA axis regulation. Early exposure to stress, such as maternal separation or child abuse, affects key structures such as the hippocampus, amygdala, and PFC, which can lead to persistent cognitive, emotional, and behavioral deficits. Animal and human models have shown that early adversity not only has immediate repercussions but can also transmit negative effects to future generations, as evidenced by epigenetic alterations and changes in brain structure [70,71]. Furthermore, the social context during childhood and adolescence plays a crucial role in mitigating or exacerbating these impacts.

A better understanding of the phenomenon of early stress, its effects, and their relationship will provide tools to develop interventions that promote resilience and reduce the vulnerabilities generated by these early experiences. In the next section, we discuss stress resilience, models for studying it, and its interaction with neuroplasticity.

## 3. Resilience

### 3.1. Delimitation of the Concept of Stress Resilience

To date, the study of adverse experiences in neuroscience has focused primarily on the impact of stress, exploring the effects of various stressors on behavior and neuronal alterations. However, psychology research has recently demonstrated that some individuals exposed to extremely adverse situations, such as physical, psychological, or sexual abuse, develop mechanisms that allow them to maintain their mental health. These mechanisms, termed “resilience mechanisms”, have become a topic of growing interest in both psychology and neuroscience.

Despite its relevance, the term “resilience” lacks a uniform definition in the scientific literature. Some authors define resilience as the ability to overcome adversity and maintain adequate physical and psychological development [2]. Others conceptualize resilience as the ability to achieve “relatively good” outcomes despite exposure to adverse experiences [1]. Aldwin (2009) broadens this perspective by suggesting that resilience is not only resistance to adverse events but also the capacity to grow or develop despite them [8]. Other authors adopt a more neutral approach in their definitions, avoiding exclusively human characterizations. For example, Karatsoreos and McEwen (2011) define resilience as the capacity of an organism to respond to environmental stressors through the efficient recruitment of allostatic responses [4]. Resilience has been described as a two-dimensional process involving both the experience of adversity and positive adaptation. In this construct, two main dimensions are evaluated: (1) the level of risk or adversity associated with negative circumstances and (2) the degree of observable positive adaptation, such as social competence or success in life [72]. For example, emotional support from caregivers during childhood can attenuate the adverse effects of early experiences, protecting cognitive and emotional development [73].

Furthermore, recent research suggests that resilience is mediated not only by genetic and neurobiological factors but also by environmental and social variables, such as social support, the quality of interpersonal relationships, and access to adequate resources [74]. This integrative approach highlights the interaction between internal and external factors that facilitate successful adaptation in the face of adversity. In the neurobiological field, resilience has been defined as an organism’s ability to respond to environmental stressors through the efficient recruitment of allostatic responses [9]. This definition underscores the role of biological mechanisms, such as regulation of the HPA axis, synaptic plasticity, and the function of key structures such as the hippocampus and amygdala in stress modulation. From our perspective, we propose a unifying vision that takes into account all the previously proposed positions. Therefore, resilience is a dynamic and multidimensional process that enables organisms, including humans, to cope with adversity through the interactions of biological, psychological, and social factors. This process not only involves the ability to resist and recover from adverse situations but also the potential for positive adaptation and, in some cases, personal growth. From a neurobiological perspective, resilience is supported by the regulation of the HPA axis, synaptic plasticity, and the function of key brain structures, such as the hippocampus and amygdala, facilitating efficient responses to environmental stressors. Thus, resilience emerges from the integration of internal and external mechanisms that promote adaptation and well-being in the face of adverse conditions.

Animal studies have provided significant evidence for understanding the mechanisms of resilience and vulnerability. For instance, experimental paradigms such as chronic social defeat stress or the forced swimming test (FST) enable the evaluation of resilient and vulnerable phenotypes in rodents. Studies employing these models have revealed that differential responses to stress are mediated by genetic, epigenetic, and neurochemical factors, including variations in the levels of neurotransmitters such as serotonin (5-HT) and brain-derived neurotrophic factor (BDNF) [75]. More details concerning the resilience study models are discussed later in this work.

Intermittent exposure to stress in early life (such as the maternal separation model [76] has been suggested to induce resilience rather than vulnerability. This hypothesis, which is based on studies by Levine et al. conducted in the 1950s, suggests that moderate levels of stress during development can promote adaptations that improve an organism’s ability to cope with future adversities [77].

Differentiating resilience from other related concepts, such as vulnerability, invulnerability, stress resistance, adaptive behaviors, and mental fortitude, is important. Vulnerability is the tendency of an organism to suffer significant psychological or physiological changes due to its interaction with external or internal events [78]. Invulnerability refers to absolute resistance to the negative consequences of adverse situations, whereas resilience implies recovery and, in some cases, growth or development after trauma [79]. For example, Carver (1998) used the term “thriving” to describe the ability to surpass the previous level of functioning after facing adversities, often resulting in more effective subsequent functioning [80]. Stress resistance refers to the ability to withstand adversity without succumbing but without necessarily improving; resilience includes stress resistance with the possibility of subsequent growth [79,80]. Recovery from adversity represents a pathway in which normal functioning is temporarily interrupted by the manifestation of psychiatric symptoms, followed by a slow recovery to the pretrauma level; in contrast, resilience implies a stable trajectory of healthy functioning over time [5]. Figure 4 illustrates our proposal of possible responses to a stress-triggering event over time according to the level of resilience that an individual possesses.

Despite the growing interest in this construct, the implementation of resilience training and monitoring programs has proven challenging due to the absence of objective measurements. Currently, resilience assessments in humans are predominantly based on psychometric tests or interviews, which, although valuable, are susceptible to subjective biases [5]. Therefore, developing effective tools based on neurobiological knowledge is very important to evaluate resilience and promote it in at-risk groups.

As we have previously described in this article, resilience is a complex phenomenon encompassing biological, cognitive, and behavioral responses to adverse events. Despite advances in its study, challenges in defining and measuring this concept remain. Understanding the interactions among genetic, epigenetic, neurobiological, and environmental factors will be key for developing effective interventions that promote resilience in vulnerable populations, especially during childhood, due to the brain and behavioral changes exerted by stress that can last throughout the life of the individual and be transmitted to subsequent generations, driving vulnerability in several family members over the years. We organized the information in a more synthetic and delimited way in Table 1 to facilitate an understanding of the different dimensions of resilience.

### 3.2. Models of Stress Resilience

Numerous stress models have been developed to study the roles of gene–environment interactions in several diseases. Recent research has shown that disease development is affected not only by stress exposure but also by individual vulnerability and resilience to that exposure. Three main approaches have been used to model vulnerability and resilience: selective breeding, selection of subpopulations, and transgenic models [31]. We describe them below.

#### 3.2.1. Selective Breeding

Selective breeding, which is based on specific phenotypes, enables the study of the influence of genetic predispositions on the stress response. Selective breeding is also known as artificial selection, a process in which specific traits of the organism are intentionally selected to be passed on to the next generation; it involves an intentional selection of individuals with specific traits from a heterogeneous population and crossing them over multiple generations to create lines with divergent phenotypes for the trait of interest [86]. In the context of stress, this technique has been used to develop animal models that exhibit extreme stress responses, allowing the study of the underlying genetic and neurobiological mechanisms of vulnerability or the capacity for recovery. This approach is a useful model to evaluate vulnerability and resilience due to the selection of specific subpopulations that can be employed to further increase such specific features of interest in the subgroups. After a few generations, the phenotype of the breeding lines diverges and allows greater characterization and investigation of the underlying molecular principles or their use as disease models.

A prominent example is the generation of mouse lines with high (HR) and low (LR) hypothalamic–pituitary–adrenal (HPA) axis reactivity. This process begins with a population of outbred mice, such as the CD-1 strain, where the individual stress response is assessed using a stress reactivity test (SRT) that measures corticosterone levels (the stress hormone in rodents) in response to a brief period of restraint. Mice showing very high or very low corticosterone secretion in the SRT are selected to establish the HR and LR breeding lines, respectively. Additionally, a third line with intermediate reactivity (IR) can be established as a control group. From the first generation, significant differences in HPA axis reactivity are observed between the HR, IR, and LR lines. These differences are maintained and can be increased in subsequent generations through selective breeding, indicating a genetic basis for the respective phenotype. Repeated SRTs indicate that these differences in stress responsiveness are present from an early age and can be considered a robust genetic predisposition [87]. Rats with high or low levels of exploratory behavior (referred to as high-responding and low-responding, respectively) have been bred using this model, and both anhedonia and anxiety have been evaluated [87,88]. For example, low-responding stressed rats took significantly longer to approach and consume a palatable treat [87], and among mice bred for CORT reactivity to an acute stressor, animals with a high response exhibited cognitive deficits and reduced hippocampal BDNF levels [89]. Numerous research reports have used this model successfully to study stress in different animals over the past few decades [90,91,92,93]. The main advantage of selective breeding as a model is that it allows the study of causality, since phenotypic differences exist before exposure to stress. Furthermore, by using genetically distinct lines, gene–environment interactions that modulate vulnerability and resilience can be investigated. However, selective breeding also has limitations. It is a high-cost and time-consuming process, requiring multiple generations to achieve robust phenotypic divergence. In addition, selectively bred lines may be susceptible to genetic drift, which can lead to the appearance of unwanted traits such as the development of other health conditions that make the communities unviable. Despite these limitations, selective breeding is a valuable tool for investigating gene–environment interactions and genetic vulnerability to disorders such as anxiety [94]. Animal models of selective breeding facilitate an exploration of the consequences of stressor experiences in relatives and their impacts over generations, as well as provide the opportunity to explore the impacts of interventions to ameliorate the effects of stress on brain circuits in order to promote adaptative behaviors to cope with adversity.

#### 3.2.2. Selection of Subpopulations

The selection of subpopulations is an approach to studying gene–environment interactions that utilizes the intrinsic heterogeneity of entire populations. In this method, one or more characteristics are investigated. The animals are grouped into subpopulations based on, for example, their performance in a behavioral test. Bergström et al. (2008) used chronic mild stress (CMS) to induce an anhedonic phenotype in rats. However, not all rats developed the phenotype; therefore, the cohort was divided into CMS-resistant and CMS-susceptible animals, allowing the identification of molecular changes between these sub-phenotypes [39]. One of the main findings was that BDNF levels are increased in the hippocampus of resistant rats. In another example of subpopulation selection, Hodes et al. (2015) reported that resilience to the development of depression-like behavioral phenotypes after exposure to social defeat stress is associated with differences in the immune system, such as a decrease in circulating proinflammatory molecules [95]. Although researchers have not yet fully clearly determined whether resilience is the cause or the consequence of these differences in the immune system, recent evidence suggests that the harmful consequences of stress may be reversed by reducing the degree of inflammation produced. Furthermore, researchers have hypothesized that circulating proinflammatory molecules released in response to chronic stress exposure penetrate the blood–brain barrier (BBB) and affect neural circuits that mediate stress vulnerability and depression [96]. Therefore, whether an organism exhibits resilience or vulnerability may depend on neurovascular health and maintaining BBB integrity to prevent stress-induced inflammatory molecules from entering mood-related brain structures [97]. This model has been complemented by extensive investigations of genetic differences [39,98], as well as noninvasive imaging techniques [99]. Despite extensive research, only single candidate genes for stress resilience persist, and the authors state that the resistant phenotype is likely caused by a combination of many different factors. Working with subpopulations offers several benefits but does not lack drawbacks. According to Scharf and Schmidt (2012), this approach can lay an excellent foundation for unbiased approaches, such as genome-wide, transcriptomic, proteomic, epigenomic, or metabolomic studies. Subpopulation selection can help identify new targets due to the selection of a phenotype as opposed to a single gene. Subpopulations are often characterized after stress exposure which prevents clear assertions about the causality of phenotypic variations in stress vulnerability [100].

#### 3.2.3. Transgenic Models

Transgenic animal models are a useful tool for the scientific community, mainly due to the refinement of gene-targeting techniques, the commercial availability of many transgenic lines, and the decrease in associated costs. Transgenic models enable studies of the roles of specific genes in different diseases and behaviors, such as vulnerability and resilience to stress. Several studies have reported and highlighted the influence of individual genes on stress vulnerability or resilience. The different types of knockout (KO) methods are described below. Conventional methods, in which a specific gene is inactivated from the entire animal genome, can provide valuable information on the general function of the specific gene; however, these methods can present many biases or specificity challenges since genes have different and even opposing functions in different tissues or cell types or during different developmental stages. Conditional KO allows for the specificity of selective gene KO in the central nervous system, individual brain regions, or specific cell types, which enables much more precise conclusions. Finally, inducible KO can be activated in a predetermined developmental period [100]. The combination of conditional KO and inducible KO with different levels of spatial and temporal specificity is the most selective and useful technique, but it is also the most complex and expensive technique. Improvements in molecular techniques such as viral vectors and optogenetics allow for the possibility of modifying genes precisely and specifically. The choice of KO type depends on the research question. Molecular techniques and optogenetics offer alternatives to traditional transgenic models. Recent research has been conducted to explore genes with already known roles in the stress response to identify new gene candidates. For instance, KO models of the pituitary glucocorticoid receptor were protected from the stress-induced increase in basal CORT levels but not from a stronger response to acute stress. Furthermore, stress exposure increased anxiety-like behaviors in wild-type (WT) animals but not in KO animals, suggesting that adverse events in early life can have beneficial effects on organismal development that improve stress-coping behaviors later in life [101]. Another interesting example is a KO model with modulated glutamate (GLU) transmission, which has also been implicated in depression. Compared with their WT littermates, vesicular glutamate transporter 1 (VGLUT1)-KO mice exposed to stress, presented a stronger anhedonic phenotype in the sucrose preference test, greater immobility in the FST, and increased ambulation. Interestingly, anxiety-like behavior and object-recognition memory were modulated by stress independent of the genotype. Notably, heterozygous KO of VGLUT1 caused a potentially compensatory increase in VGLUT2 levels in the frontal cortex and hippocampus. The authors highlighted the crucial role of decreased VGLUT1 levels in the forebrain as a biological mediator of increased vulnerability to CMS [102].

Collectively, these three methodological approaches, each with their own advantages and limitations, provide valuable insights into the complex interactions between genes and the environment in modulating the stress response and their implications for health and disease. The most appropriate model will depend on the specific research question, seeking a balance between specificity, causality, and experimental feasibility. Continued improvements in molecular biology techniques, such as optogenetics and viral vectors, also offer new possibilities for precise and temporally controlled genetic manipulation, complementing and expanding the capabilities of traditional models.

## 4. Neuroanatomical Structures and Plasticity Mechanisms Involved in Stress and Resilience Responses

### 4.1. Reward Circuit

The majority of the evidence linking specific brain structures to stress resilience, particularly in murine models, has positioned the reward system and its components as fundamental components of regulatory mechanisms. We present a summary of these components and their functions to provide a better understanding of how the reward system and its components participate in this process. The mesolimbic pathway has been widely recognized as the brain center that controls reward and motivational mechanisms (see [103,104,105]), regulating motivated behaviors such as eating, sex, and sleep. This pathway is also activated by the consumption of substances of abuse and by situations, stimuli, or behaviors that can be considered pleasurable [104,106,107]. The mesolimbic system, particularly the mesolimbic–cortical system, includes structures of the midbrain, the limbic system, and the neocortex, which are closely interconnected. In turn, several neurotransmission systems participate in the interconnection of these structures and regulate the acquisition and maintenance of motivated behavior or substance addiction. The main neurotransmitters include dopamine (DA), opioid peptides (OPIs), gamma-aminobutyric acid (GABA), GLU, 5-HT, and norepinephrine (NA) [108]. One of the main neurotransmission pathways in this system originates from the midbrain, specifically the ventral tegmental area (VTA), which projects to several limbic and cortical nuclei, such as the nucleus accumbens (NAc), amygdala, hippocampus, hypothalamus, and PFC, among others, and is a dopaminergic region [109]. This pathway is required for the initiation and maintenance of motivated behaviors, such as drug use, palatable food intake, or sex, through the regulation of the rewarding effects of these behaviors, similar to the consumption of drugs such as cocaine [110]. Both the consumption of drugs of abuse and highly palatable foods and addictive behaviors such as sex produce an increase in DA levels derived from projections from the VTA to limbic–cortical structures [111]. DA acts on the limbic system, particularly on the NAc and amygdala, which are some of the most important structures involved in emotional processing and stimulus valence assignment [112]. Some dopaminergic projections terminate in the PFC, particularly in the dorsolateral, ventromedial, and orbitofrontal PFC [113].

Motivated behaviors depend on the reward system due to its relevance to the maintenance of biological system homeostasis. Reinforced behaviors ensure both group and individual survival. Some primary motivated behaviors are sex, hunger, and thirst, which are necessary for the survival of the individual or the species. On the other hand, sex seems to have no relevance to the survival of the individual; however, it is necessary for reproduction in species that use sexual reproduction. Otherwise, the survival of the entire species is endangered. Motivated behaviors undergo two sequential phases, anticipatory and consummatory. The anticipatory phase is characterized by increasing desire for a reward (food, water, or sex) and the initiation of motor patterns (e.g., walking) to approach or seek a gratifying or rewarding stimulus; in addition, this phase is accompanied by other processes, such as the search for and choice of food or the search for and courtship of a sexual partner. This phase depends on DA, which is released in the brain at the moment of seeking a reward [114,115]. DA seems to mediate the search to achieve what is desired. In contrast, the consummatory phase involves the moment of consumption or reaching the rewarding element and the subsequent sensations. This phase is known as the ‘liking’ phase [116] because it is precisely related to the liking of a reward. Namely, during this phase, pleasurable associations will be generated that allow organisms to remember things that generate well-being and move away from things that generate displeasure. This phase is associated with the release of OPIs, with data showing that central nervous system opioids play an important role in hedonic pleasure when eating [117]. In the consummatory phase, the release of OPIs increases and that of DA decreases; specifically, opioids are released from the beginning of the consummatory phase until shortly after the end of the phase [115,118]. Notably, the relationship between the two main neurotransmitters (DA and opioids) is not linear but rather circular, as DA facilitates the release of opioids, which inhibit DA release [119]. In this case, DA causes an increase in motivation and searching behavior, which results in an episode of hedonic pleasure due to the reinforcing effect mediated by opioids. In turn, opioids inhibit DA, momentarily stopping the desire and search for food that generates pleasure. This pattern explains the refractory periods observed in both eating and sexual behavior. Understanding the bearing of the reward system and the regulation of motivated behaviors allows researchers to better comprehend the impact of its participation in the stress response and the development of resilience.

### 4.2. Reward Circuit, Stress, and Resilience

Functional and structural changes in brain circuits associated with emotional regulation, especially those involving the PFC and amygdala, have been implicated in resilience. Resilient individuals show greater connectivity between the PFC and the amygdala, suggesting better top-down regulation of emotional responses. Furthermore, adaptive plasticity in the VTA and NAc pathways has been linked to the modulation of reward and motivation systems, which could counteract stress-induced anhedonia.

Using a validated CMS model to induce depression, Febbraro et al. (2017) suggested that changes in neuronal activity are associated with CMS endophenotypes that are resistant or susceptible to stress [120]. An examination of the changes in c-Fos expression in 13 brain regions associated with stressful conditions revealed that six regions, all of which are limbic structures, are sensitive to stress exposure. CMS was found to suppress the c-Fos response within the magnocellular ventral lateral geniculate nucleus in both stress subgroups. In the lateral and ventral orbital cortices of resilient rats, c-Fos immunoreactivity was also suppressed by stress exposure. Conversely, the c-Fos response within the amygdala, medial habenula, and infralimbic cortex was selectively increased in stress-susceptible rats. The effects within these regions are associated with the hedonic state of the rats. Therefore, these regions could be associated with the stress-coping mechanisms underlying CMS-induced segregation into stress susceptibility and stress resistance.

Anacker et al. (2016) examined synchronized anatomical differences between brain regions to gain insights into the plasticity of neural networks underlying stress susceptibility. C57BL/6 mice were exposed to 10 days of social defeat stress and subsequently evaluated using social avoidance tests [121]. Using magnetic resonance imaging, the brains of stressed mice (susceptible, resilient, and control) were imaged in vivo. Social avoidance was negatively correlated with the local volume of the cingulate cortex, NAc, thalamus, raphe nuclei, and bed nucleus of the stria terminalis and positively correlated with the volume of the VTA, habenula, periaqueductal gray, cerebellum, hypothalamus, and hippocampal CA3. Synchronized anatomical differences were observed between the VTA and cingulate cortex, the hippocampus and VTA, the hippocampus and cingulate cortex, and the hippocampus and hypothalamus. These correlations revealed different structural covariances between brain regions in susceptible and resilient mice. Figure 5 presents a general scheme of the main neurotransmission systems and cortical mesolimbic structures of the human brain involved in the reward circuit, stress, and resilience.

β-Catenin is a multifunctional protein that plays an important role in the mature central nervous system; dysfunction of this protein has been implicated in various neuropsychiatric disorders, including depression. In mice, β-catenin mediates anxiolytic and stress resilience-promoting effects in the NAc, a key brain reward region, an effect that is mediated by D2-type medium spiny neurons [122].

Tang et al. (2019) analyzed differential changes in the proteome by combining mass spectrometry and isobaric tags for relative and absolute quantitation (iTRAQ) [122]. Among the 2593 quantified proteins, 367 were aberrantly expressed [123]. These hippocampal protein candidates could be associated with stress-induced depression or anxiety susceptibility and stress resilience. These findings identify potential protein systems involved in various metabolic pathways as new research targets. Furthermore, an independent immunoblot analysis revealed changes in the levels of several proteins which were specifically associated with the depression-susceptible, anxiety-susceptible, and nonsusceptible groups, respectively, suggesting that identical CMS differentially affected metabolic and mitochondrial processes in the hippocampus. Taken together, the observed alterations in the hippocampal protein abundance profiles provided significant and novel insights into the mechanism of stress regulation in a CMS rat model [123]. Table 2 summarizes the main brain structures involved in the processes of motivation, addiction, stress response, and resilience based on the scientific literature. For each structure, its specific functions within these domains are detailed, highlighting how they participate in the perception, evaluation, and response to stimuli, as well as in the adaptation to adversity. The table seeks to provide a synoptic view of the complex neuronal network underlying these important aspects of behavior and mental health.

### 4.3. Plasticity Mechanisms Associated with Stress and Resilience

Neuroplasticity is defined as the capacity of the nervous system to adapt its activity, connectivity, or morphology in response to internal or external stimuli through a structural reorganization of functions and/or networks [137]. These changes can be favorable for the brain, such as a regenerative process after injury; neutral, where no consequences are observed after the adaptive process; or negative, leading to a pathological state [138]. Brain adaptations observed after stressful experiences in early life can be classified in the last category. Maternal separation is a murine paradigm that is employed to study the effect of adverse experiences at early ages on brain plasticity. The maternal separation model is a widely used tool for modeling early-life adversity. The impact on the hippocampal cytoarchitecture due to maternal separation can be observed when separation occurs intermittently. Separating pups from their mothers for 15 min daily on postnatal days 2 to 14 causes alterations in the morphology of cells in the hippocampal CA3 region, as well as reduced exploratory activity and decreased short-term memory in the exposed pups when they reach adulthood [125]. Prolonged early stress, when pups are separated from their mother for 3 h daily from postnatal days 2 to 14, impacts their maternal behavior toward their own pups, resulting in less caring behaviors. This paradigm of maternal separation impacts the next generation. In the study, the memory of pups born to females that were stressed during the early postnatal period was evaluated, revealing poor performance in navigation memory [125]. The harmful effects of stress during the early stages persist into aging. In a study using the paradigm of 3 h of separation daily from postnatal days 2 to 14, electrophysiological recordings were performed at 70 weeks of life, which revealed alterations in long-term potentiation in the CA1 and CA3 hippocampal regions of Wistar rats, highlighting the long-term impact of early adverse events [124]. Plastic changes induced by intermittent maternal separation combined with early weaning are related to decreased densities of parvalbumin and somatostatin interneurons in the ventral hippocampus. These two types of interneurons are important because their balance is essential for the proper functioning of theta oscillations in the hippocampus. The study also revealed hyperactivation of theta power when recorded with electrodes implanted in the ventral hippocampus during exposure to novel environments. Theta brain waves are patterns of electrical activity in the brain that oscillate in the frequency range of 4 to 8 Hz. These waves are associated with various cognitive and behavioral functions and mental states such as memory, learning, and spatial navigation [139]. An overactivation of theta waves refers to an abnormal increase in the amplitude or frequency of these oscillations. An excessive or dysregulated overactivation of theta waves is associated with impaired cognitive and emotional functions [41,124] These increased activity levels might suggest heightened arousal or altered emotional responses in these rats, which could indicate dysregulation of the ventral hippocampal circuitry that regulates anxiety behaviors under regular conditions [41]. Adult hippocampal neurogenesis (AHN) is one of the neuroplasticity events involved in the anxiety response, and it is a process that results in the generation of new neurons that form an in situ precursor cell population through a series of developmental steps [140]. This phenomenon occurs in specific areas of the brain, one of which is the hippocampus. In models of ELS induced by maternal separation, the hippocampi of rats separated from their mothers at an early age exhibit reduced proliferation of new neurons in the dentate gyrus and a decrease in the number of cells that acquire neuronal identity when they reach adulthood [127]. Interestingly, Murthy et al. (2019) did not detect differences in neurogenesis in the ventral hippocampus between adult rats that were exposed to maternal separation and early weaning and adult rats that were exposed to early stress as pups [41]. Modeling early-life adversity is not restricted to the maternal separation paradigm. Limited nesting and bedding material also induces adaptations in cerebral features. Limiting nesting material from postnatal days 2 to 9 has lasting effects on hippocampal structure and function. In male mice, the survival of new neurons in adulthood is reduced, which is correlated with poor performance in the object recognition task and Morris water maze. The Morris water maze learning paradigm is a widely used behavioral test in rodent research (primarily in rats and mice) to assess hippocampus-dependent spatial learning and memory. It is based on the animals’ ability to learn and remember the location of a hidden platform in a pool filled with opaque water using spatial cues from the environment [129]. Interestingly, the females in the study exhibited less of an impact of the chronic stress experienced at an early age [131]. Considering this observation, the same research group evaluated the impact of the exposure of females to early stress on the modulation of adult neurogenesis at 8 months of age via exercise. Several conditions can increase adult neurogenesis, such as environmental enrichment, sexual activity, and voluntary exercise [126]. Abbink et al. (2017) evaluated female mice running for six weeks and reported that female mice exposed to early stress had less neurogenesis than animals of the same age that did not experience early stress, revealing a lasting stress effect on the responsiveness of the neurogenic niche [42]. One mechanism underlying the effects of ELS on poor cognitive performance and hippocampal dysfunction is the BDNF–TrkB pathway [128].

Increased neurogenesis is associated with greater cognitive flexibility and emotional regulation, mediating the effects of stress on the hippocampus. Studies of rodents have shown that increasing neurogenesis through environmental enrichment or pharmacological agents can promote resilience to stress-induced depressive behaviors. In young rats, intermittent exposure to an enriched environment decreases the expression of anxiety markers and increases the number of new neurons and the complexity of their connections [141]. Neurogenesis in the adult hippocampus, particularly the dentate gyrus, has been strongly implicated in resilience [142].

Glucocorticoids, hormones released in response to stress, exert complex and multifaceted effects on AHN. A mechanism of action is the modulation of glutamatergic neurotransmission, specifically through increased GLU release and the potentiation of NMDA receptor-dependent excitatory inputs from the entorhinal cortex to newly generated neurons [143]. However, this effect is not uniform; glucocorticoids can also have proapoptotic effects on neural progenitor cells (NPCs) and immature neurons in the hippocampus [144], potentially leading to NPC depletion and, therefore, a decrease in neurogenesis. This duality in glucocorticoid action underscores the importance of the context and dose in determining the net effect of glucocorticoids on neurogenesis. Recent studies suggest that chronic exposure to glucocorticoids, but not acute exposure, negatively affects NPC proliferation and differentiation in the dentate gyrus [13].

In addition to stress hormones, other neurotrophic factors play crucial roles in regulating hippocampal neurogenesis. These factors include BDNF, the mitogen fibroblast growth factor 2 (FGF-2), vascular endothelial growth factor (VEGF), and insulin-like growth factor 1 (IGF-1), which promote cell proliferation and differentiation in the dentate gyrus and mediate some of the positive effects of enriched environments [145,146,147]. In particular, BDNF has been linked to synaptic plasticity and neuronal survival, playing a key role in the antidepressant effects of various treatments [148]. BDNF plays an important role in regulating synapse formation and the maturation of new neurons, plasticity, and dendritic arborization [149]. VEGF exerts angiogenic and neurotrophic effects, contributing to the vascularization of the neurogenic niche and the support of new neurons [150]. IGF-1 is related to neurogenesis and neuronal protection, exhibiting complex interactions with the glucocorticoid system that favor neurogenesis and synaptogenesis [151].

Gratifying social experiences also influence neurogenesis through the release of neuropeptides such as endogenous opioids and oxytocin, as well as the neuromodulator DA [129,152]. Endogenous opioids activate opioid receptors and modulate synaptic plasticity and neurogenesis in response to rewarding stimuli. Sustained activation of the μ opioid receptor decreases structural plasticity and neurogenesis in the hippocampus [153]. Oxytocin, known for its role in social bonding and emotional regulation, is also associated with an increase in hippocampal neurogenesis, suggesting that oxytocin may protect against the suppressive effects of stress hormones on hippocampal plasticity and stimulate neuronal growth [130]. The innervation of DAergic neurons can also regulate neurogenesis in the hippocampus [154].

One structure that has attracted considerable attention in recent years is the habenula. The habenula is a bilateral brain structure that is a component of the dorsal diencephalic conduction system [134], which comprises the stria medullaris (SM), fasciculus retroflexus (FR), and habenular nucleus (Hb) [155]. Its name originates from the Latin “habena”, meaning “little rein”, a reference to its distinctive morphology. Specifically, the lateral habenula (LHb) receives afferent projections from the basal ganglia, hypothalamus, and limbic regions, integrating information concerning both the organism’s internal state and the current external environment. This integrated information is then conveyed to various brainstem regions by ascending projections, contributing to the modulation and updating of behavior to facilitate adaptation in a dynamic environment. Consequently, the LHb plays a crucial role in learning to inhibit specific responses to distinct stimuli [136]. Furthermore, research suggests a link between neurotrophic factors and habenular function. The authors proposed that increased BDNF levels contribute to increased cell proliferation within the medial habenula, as documented by Pencea et al. (2001), who reported that BDNF promoted cell proliferation and neurogenesis in both the hypothalamus and the habenula [132]. Given the role of the habenula in mediating responses to stressful and aversive stimuli [135], Sachs and Caron (2015) suggested that neurogenesis within this structure may be important for buffering stress responses and mediating behavioral responses to the antidepressant fluoxetine [133].

When adverse situations occur, mechanisms involving the immune, endocrine, and nervous systems are activated. These mechanisms are necessary for neuroplasticity, which prepares organisms to respond to learned environmental contingencies or future threats [15,40]. Psychosocial adversity affects and calibrates the related biological systems and depends largely on the organism’s developmental stage and the sources and nature of the environmental inputs that shape the brain. Furthermore, the plasticity mechanisms change throughout development [40]. Human brain development is a long process that begins during gestation and lasts at least two decades. This process involves the following neuroplastic mechanisms: Neurogenesis, which is defined as the process by which new neurons are formed [156], occurs mostly during embryonic development. Synaptogenesis, which involves the formation of synaptic connections [157,158] and occurs to a greater extent during the first year of life, reaching its peak between 2 and 4 years of age, as well as during adolescence; however, this process can occur at any time in life. Synaptic pruning is a developmental process in which the elements that comprise a genuine structural synapse (presynaptic terminals and postsynaptic membranes) are eliminated and the elimination of small segments of axonal and dendritic branches can occur [159]; this process reaches its peak between 2 and 10 years of age and helps to consolidate synaptic connections. Gliogenesis, the birth of new glial cells [157], occurs through synaptic remodeling and myelination.

As previously described, the allostatic load refers to the consequences of chronic activation of the stress response, including the HPA axis and glucocorticoid secretion, which in turn causes endocrine, immunological, and neural adaptations [14]. Although short-term physiological changes allow organisms to maintain homeostasis in the face of adverse situations, over time, these adaptive responses lead to wear and tear of regulatory systems. For example, acute exposure to threatening situations triggers the secretion of cortisol, excitatory amino acids, and inflammatory cytokines, which generate changes in the hippocampus, such as dendrite retraction and a reduction in synaptic density. These changes have been linked to the development of anxiety and depression [6].

Chronic immobilization stress causes different changes in the amygdala. For example, an expansion of dendrites is observed in the basolateral amygdala, whereas the downregulation of dendritic spines occurs in the medial amygdala, which enhances fear learning in response to similar events [6,40]. These changes in the hippocampus and amygdala have been linked to memory problems and other behavioral and cognitive symptoms that are common in individuals with depression [40].

Adult neurogenesis not only is required for cognitive and emotional functions but also acts as a pillar of brain plasticity, modulating stress responses and behavioral adaptations in a regional manner, with dorsal hippocampal neurogenesis associated with cognitive performance and ventral hippocampal neurogenesis related to emotional regulation. These findings highlight the importance of promoting lifestyles and environments that favor this process and suggest new opportunities for the development of therapies aimed at neuropsychiatric and aging-related disorders.

## 5. Conclusions

The relationship between stress and resilience represents a dynamic and multifaceted process influenced by numerous biological, social, and environmental factors. Throughout life, organisms encounter various stressful events whose nature and magnitude can determine their capacity to adapt or succumb to stress. This process is mediated by a complex network of neuroplastic mechanisms, in which neurotrophic factors such as BDNF and transcription factors like c-Fos play fundamental roles. These plastic mechanisms can facilitate adaptation in certain circumstances while contributing to dysfunction in others, particularly when stress becomes chronic. Chronic stress negatively impacts brain plasticity, especially in key limbic regions, including the hippocampus, amygdala, orbitofrontal cortex, and habenula, potentially triggering alterations in adaptive capacity and emotional regulation. Furthermore, the stress response and resilience are intimately linked to the modulation of various signaling pathways involving neurotransmitters such as monoamines, amino acids, and opioids—all of which are essential for regulating emotional, cognitive, and behavioral states. In this context, the reward system critically influences the hedonic component, while the defensive system, by activating immune responses, promotes the release of pro-inflammatory factors, generating alterations that may compromise long-term health.

Although various experimental models have been developed at both the molecular and behavioral levels to study these phenomena, their specificity limits the integration of complex resilience and stress responses in naturalistic contexts. Nevertheless, multiple protective factors have been identified, including enriched environments and safe, stimulating social interactions, which promote adaptation processes and mitigate the adverse effects of stress, thereby enhancing individual resilience. Knowledge of the mechanisms underlying stress and resilience continues to advance, yet we must acknowledge that these phenomena result from interactions between biological, social, cultural, and environmental factors. This multidimensional approach is essential for developing more effective intervention models aimed at fostering healthy contexts and habits that can prevent the dysfunctional and pathological effects of stress. Progressing toward a more comprehensive understanding of these processes will enable the design of more precise strategies to mitigate the impacts of stress and promote resilience across different contexts, contributing to overall well-being.

Finally, adverse experiences during early developmental periods may alter brain function, with changes potentially persisting throughout life. Resilience has a clear biological foundation, and understanding these mechanisms creates opportunities to develop timely interventions that promote resilience as a neuroprotective factor for susceptible populations. Such interventions have the potential to improve the quality of life of current and future generations.

## Figures and Tables

**Figure 2 ijms-26-03028-f002:**
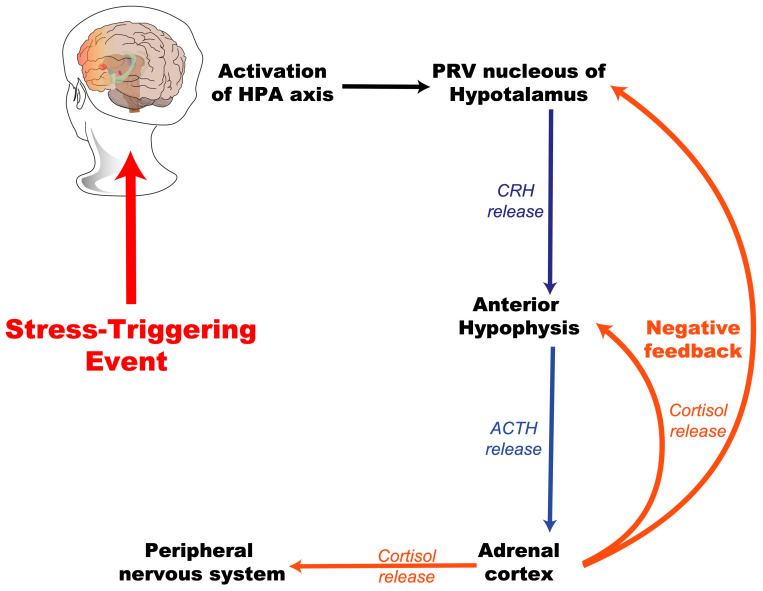
Hypothalamic–pituitary–adrenal (HPA) axis activation. This diagram illustrates the activation of the hypothalamic–pituitary–adrenal (HPA) axis in response to a stress-triggering event. The HPA axis is a crucial neuroendocrine system that mediates the body’s stress response. The HPA axis is activated when a stressor is perceived. This activation starts in the hypothalamus (PRV nucleus), which releases CRH. CRH prompts the anterior pituitary to release ACTH. ACTH then stimulates the adrenal cortex to release cortisol. Cortisol, the stress hormone, circulates in the body, triggering the stress response and providing negative feedback to the hypothalamus and pituitary to regulate the system. PRV, paraventricular nucleus of the hypothalamus; CRH, corticotropin-releasing hormone; ACTH, adrenocorticotropic hormone.

**Figure 3 ijms-26-03028-f003:**
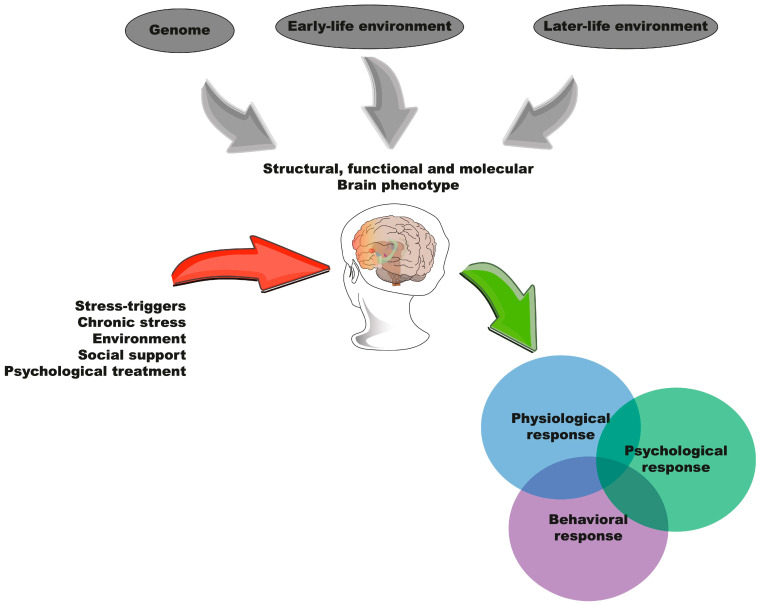
Integrated model of early-life stress and its long-term effects. The integration of the three different hypotheses that have been proposed to explain the outcomes associated with early-life stress and the possible psychological, physiological, and behavioral responses of individuals in adulthood.

**Figure 4 ijms-26-03028-f004:**
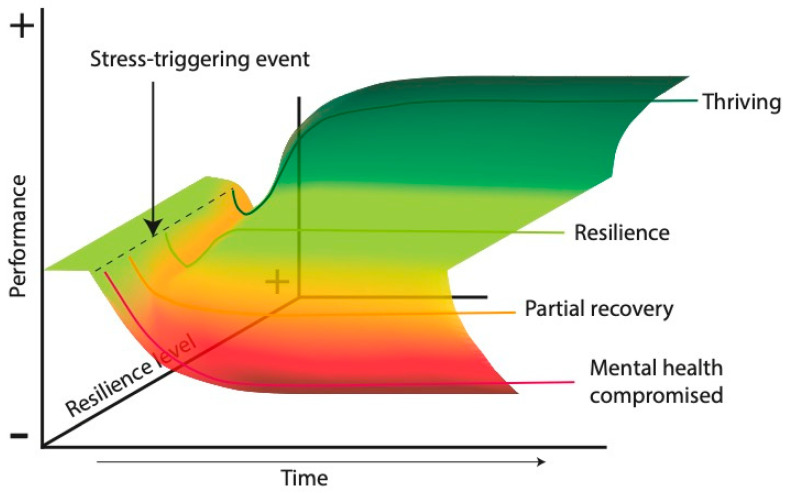
Resilience and stress event outcome trajectories. This figure illustrates the potential impact of a stress-triggering event on the performance of an individual over time, differentiated by their level of resilience. Time (X-axis): represents the progression of time following the stress-triggering event. Performance (Y-axis): indicates the level of an individual’s functioning or effectiveness, ranging from negative (−) to positive (+). Resilience level (Z-axis): depicts the ability of an individual to adapt and recover from stress, ranging from low (−) to high (+). The colored lines represent different potential outcome trajectories. Green (“thriving”) represents individuals with high resilience who not only recover but experience growth and improved performance after the stressor. Light green (“resilience”) represents individuals with moderate resilience who recover to their prestressor performance level over time. Orange (“partial recovery”) illustrates individuals with lower resilience who experience some recovery in performance but may not fully return to their previous level. Red (“mental health compromised”) depicts individuals with low resilience, where the stress-triggering event leads to a significant and prolonged decline in performance, indicating compromised mental health.

**Figure 5 ijms-26-03028-f005:**
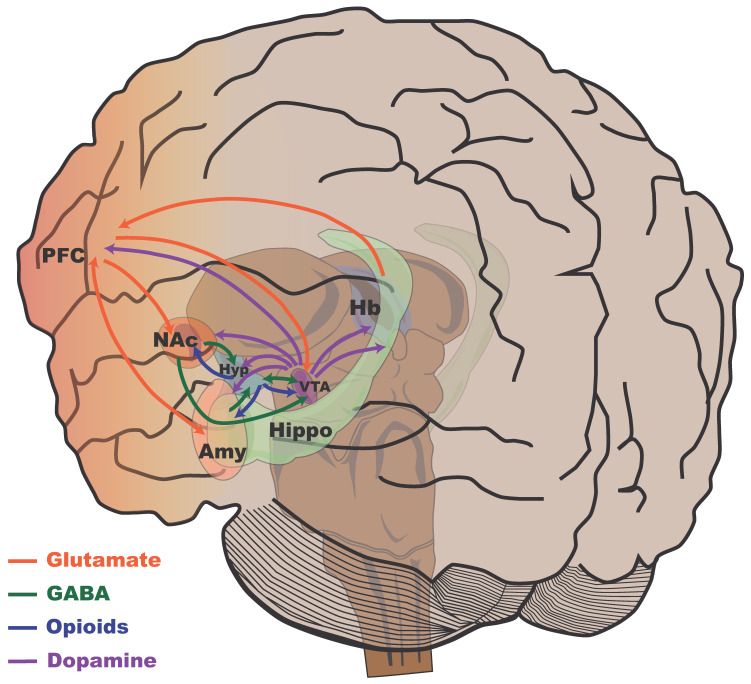
Schematic representation of the major mesocorticolimbic pathways in the human brain. This diagram illustrates key neural circuits involved in reward, motivation, and emotion, highlighting different neurotransmitter systems within these pathways in the human brain. Dopaminergic pathways (purple), glutamatergic pathways (orange), GABAergic pathways (green), and opioid pathways (blue) are shown. PFC, prefrontal cortex, which is involved in executive functions, planning, and decision-making; NAc, nucleus accumbens, a key area in the reward circuitry associated with pleasure and motivation; Hyp, hypothalamus, which regulates various bodily functions, including the stress response and homeostasis; VTA, ventral tegmental area, which is the origin of the mesocorticolimbic dopamine pathway that is crucial for reward and motivation; Hb, habenula, which is involved in processing negative feedback and aversion; Amy, amygdala, which plays a central role in processing emotions, particularly fear and anxiety; and Hippo, hippocampus, which is essential for memory formation and spatial navigation.

**Table 1 ijms-26-03028-t001:** Dimensions that comprise resilience.

Dimension	Description	Refs.
General Definition	Multifaceted capacity of an organism or individual to adapt, recover, and/or grow after adversity.	[8,72,80]
Essential Components	Significant risk: exposure to major stressors threatening well-being. Positive adjustment: adaptive response and/or growth following adversity.	[5,72]
Perspectives of the Concept	Resilience as recovery: return to the preadversity state. Resilience as growth: surpassing the previous level of functioning. Resilience as an adaptive process: dynamic interactions between internal and external factors.	[8,79,80]
Modulating Factors	Neurobiological factors: efficient regulation of the HPA axis, synaptic plasticity, and allostatic responses. Genetic and epigenetic factors: gene expression is influenced by stress experiences. Environmental and social factors: social support, relationship quality, and resource availability.	[9,74,81]
Conceptual Differentiation	Invulnerability: absolute resistance (not synonymous with resilience). Stress resistance: coping without necessarily improving. Mental toughness: psychological competency in facing challenges.	[79,80]
Bidimensional Aspect	Significant risk: magnitude of the adverse event. Positive adjustment: functional positive outcome in social or behavioral contexts.	[72,78]
Behavioral Indicators	Social avoidance, anhedonia, academic or social success, controlled emotional responses.	[78,81]
Measurement and Evaluation	Currently based on psychometric scales and subjective reports, with a need for objective methods (e.g., biomarkers).	[5,77]
Animal Models	Maternal separation Chronic social defeat stress Forced swimming tests	[76,81,82,83]
Challenges and Limitations	Lack of a unified definition. Limited standardization of evaluation methods. Inadequate use of the term in studies.	[5,74]
Gene–Environment Interaction	The social environment and genetic predisposition work together to shape resilience responses.	[84,85]
Future Applications	Development of personalized interventions to foster resilience. Creation of biomarkers for an objective evaluation. Inclusion of interdisciplinary approaches in its study.	[74,78]

**Table 2 ijms-26-03028-t002:** Key brain structures and their functions in motivation, addiction, stress, and resilience.

Brain Structure	Synthesized Functions	Refs.
Reward System	Controls motivation and pleasure, activated by natural rewards and drugs of abuse. Plasticity in the VTA and NAc modulates reward and counteracts anhedonia from stress.	[103,104,105]
Ventral Tegmental Area (VTA)	Origin of dopamine for reward and motivation. Projects to limbic and cortical regions. Its volume correlates with post-stress social avoidance.	[103,104,105,112]
Nucleus Accumbens (NAc)	Key for reward and motivation, receives dopamine signals from the VTA. Its plasticity counteracts anhedonia. Its volume is inversely related to social avoidance. β-Catenin promotes stress resilience.	[121,122]
Amygdala	Processes emotional responses and stimulus valence. Greater connectivity with the PFC in resilient individuals (better emotional regulation). Its activity increases in chronic stress susceptibility. Dendritic changes due to chronic stress affect fear learning.	[6,40,120,121]
Hippocampus	Involved in memory and navigation. Its plasticity is related to reward. Its activity is suppressed in chronic stress susceptibility. Its volume correlates with social avoidance. Early stress alters its structure and function (memory, maternal behavior, reduced neurogenesis). Adult neurogenesis influences anxiety and resilience, which is regulated by neurotrophic factors and social experiences. Chronic juvenile stress causes atrophy.	[6,42,120,121,124,125,126,127,128,129,130,131]
Prefrontal Cortex (PFC)	Involved in executive functions and planning. Greater connectivity with the amygdala is observed in resilient individuals (better emotional regulation). It receives dopamine signals. Its activity is suppressed in individuals with chronic stress resilience and increased in susceptible individuals in specific areas. Its volume is related to social avoidance. Chronic juvenile stress causes atrophy.	[6,40,113,120,121]
Habenula	Processes negative feedback and aversion. Its volume correlates with social avoidance. BDNF influences cell proliferation in this region. Neurogenesis in this area might buffer stress responses.	[121,132,133,134,135,136]
Hypothalamus	Regulates the stress response and homeostasis. Its volume correlates with social avoidance. The hippocampus relates to it differently in stress resilience and susceptibility. BDNF may promote neurogenesis in this area.	[121,132]

## Data Availability

No new data were created or analyzed in this study. Data sharing is not applicable to this article.

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
