# Peer review of "A Comprehensive Overview of Stress, Resilience, and Neuroplasticity Mechanisms"

_ijms, 2025, doi:10.3390/ijms26073028_

Round 1
Reviewer 1 Report
Comments and Suggestions for Authors
The Article titled " A Comprehensive Overview Of Stress, Resilience And Neuroplastic Mechanisms" is an interesting work, but major information missed to conclude.
- The introduction needs to be revised because you discussed the different definitions of resilience, but you repeat the same thing below in the "Resilience" section.
- You have cited the five main components of stress, a key term in the review. For this reason, you are encouraged to elaborate further on these points.
- Figure 1 : Add legend.
- The match/mismatch hypothesis and the cumulative stress hypothesis are the primary hypotheses used to explain the outcomes of ELS. For this reason, these two hypotheses should be further developed, and the results of relevant studies should be cited.
- In the section « Resilience » : Since this is a comprehensive review, you should include your opinion on these definitions and indicate which definition is the most suitable.
- Figure 2 : Write a short title with a detailed legend.
- Figure 3 : Write a short title with a detailed legend.
- Three main approaches have been used to model vulnerability and resilience: selective breeding, selection of subpopulations, and transgenic models. It would be better to cite these three approaches as subgroups.
- Develop the section « Selective breeding » and add references.
- From line 399 to 340 : Expand further on your analysis of the three different approaches.
- Figure 4 : Write a short title with a detailed legend.
- You talked a lot about pathways, but you did not explain them at the molecular level to better understand the mechanisms.
- In the review, it would be better to add a summary and complementary tables and/or figure to guide the reader.
- The topic of the review involves several areas of the brain, so it would be better to add a section on the brain anatomy, citing the different regions involved in stress and its management. In addition, it would be better to add a figure of the brain.
- The type of the review is a comprehensive review; unfortunately, we have not seen the authors' input. When citing the results of the different studies, you should provide your scientific perspective by analyzing all of this.
- In the review, you wrote long paragraphs rich in information but poor in references, which would help to support your information.
- Develop the conclusion part.
- Check reference 1; the article's date is missing.
Author Response
General statements about the revised manuscript
We have strived to address most of the reviewers' comments. We addressed those comments that we felt were the most relevant and those that were relevant to the focus of our article. In other cases, we justify the reasons for noncompliance in the point-by-point response to comments section of this letter (see below). We also performed minor modifications throughout the article in addition to those kindly recommended by the reviewers to improve clarity and accuracy. All the authors agreed with the changes in the manuscript. All the revisions/modifications of the manuscript were highlighted using the MS Word “Track Changes” tool for easy identification. An academic English editing service was hired to revise and proofread the manuscript. All the changes in the manuscript made by both the editor and us are highlighted. Below, you will find point-by-point responses to your comments. Regular font style is used for featured comments, and italics have been used to show our responses.
Point-by-Point Responses to the Reviewers’ Comments
REVIEWER 1:
The Article titled " A Comprehensive Overview Of Stress, Resilience And Neuroplastic Mechanisms" is an interesting work, but major information missed to conclude.
RESPONSE: We greatly appreciate the feedback received from the reviewer. We rewrote many parts of the article and added more than 20 additional references. Due to the extensive nature of the changes, we kindly asked the reviewer to check the changes made in the manuscript with the "Track Changes" feature activated in Word.
The introduction needs to be revised because you discussed the different definitions of resilience, but you repeat the same thing below in the "Resilience" section.
RESPONSE: This issue has been resolved. We have rewritten the Introduction section to eliminate the redundant information.
You have cited the five main components of stress, a key term in the review. For this reason, you are encouraged to elaborate further on these points.
RESPONSE: We added a more detailed description of each of the above points to the text.
Figure 1 : Add legend.
RESPONSE: This issue was resolved.
The match/mismatch hypothesis and the cumulative stress hypothesis are the primary hypotheses used to explain the outcomes of ELS. For this reason, these two hypotheses should be further developed, and the results of relevant studies should be cited.
RESPONSE: This issue was resolved. We developed both sections, adding new data from relevant studies.
In the section « Resilience » : Since this is a comprehensive review, you should include your opinion on these definitions and indicate which definition is the most suitable.
RESPONSE: This issue was resolved. We have added a unified proposal for the definition of resilience.
Figure 2 : Write a short title with a detailed legend.
RESPONSE: This issue was resolved.
Figure 3 : Write a short title with a detailed legend.
RESPONSE: This issue was resolved.
Three main approaches have been used to model vulnerability and resilience: selective breeding, selection of subpopulations, and transgenic models. It would be better to cite these three approaches as subgroups.
RESPONSE: This issue was resolved. We have developed this section.
Develop the section « Selective breeding » and add references.
RESPONSE: This issue was resolved. We have developed this section by adding several new references to the article.
From line 399 to 340 : Expand further on your analysis of the three different approaches.
RESPONSE: This issue was resolved. We have developed this section.
Figure 4 : Write a short title with a detailed legend.
RESPONSE: This issue was resolved.
You talked a lot about pathways, but you did not explain them at the molecular level to better understand the mechanisms.
RESPONSE: We agree with the need to delve deeper into the molecular mechanisms underlying the phenomena addressed in this review; however, due to the extent and level of depth needed, we believe this topic must almost exclusively be addressed in an additional publication.
In the review, it would be better to add a summary and complementary tables and/or figure to guide the reader.
RESPONSE: We added two tables, one containing the dimensions that comprise resilience and another with the key brain structures and their functions in motivation, addiction, stress, and resilience.
The topic of the review involves several areas of the brain, so it would be better to add a section on the brain anatomy, citing the different regions involved in stress and its management. In addition, it would be better to add a figure of the brain.
RESPONSE: We added a table with the key brain structures and their functions in motivation, addiction, stress, and resilience. Additionally, we already have shown different regions involved in the circuitry and the description in Figure 4. We cite other articles on the role of stress in modulating brain anatomy.
The type of the review is a comprehensive review; unfortunately, we have not seen the authors' input. When citing the results of the different studies, you should provide your scientific perspective by analyzing all of this.
RESPONSE: We have rewritten many parts of the article and added many additional references, as well as many personal interpretations and opinions from the authors. Due to the large number of changes, we ask the reviewer to check the changes made in the manuscript with the "Track Changes" option activated in Word.
In the review, you wrote long paragraphs rich in information but poor in references, which would help to support your information.
RESPONSE: We rewrote many parts of the article and added more than 20 additional references. Due to the extensive nature of the changes, we kindly ask the reviewer to check the changes made in the manuscript with the "Track Changes" option activated in Word.
Develop the conclusion part.
RESPONSE: This issue was resolved. We added a comprehensive Conclusions section.
Check reference 1; the article's date is missing.
RESPONSE: This issue was resolved.
Reviewer 2 Report
Comments and Suggestions for Authors
This is a well-researched highly informative text reviewing neurobiological basis of stress and resilience. the level of information is appropriate and up to date. Overall, the narrative is easy to follow. However, I detected certain inconsistencies in the level of communication throughout the review. In some parts the information is rather basic (e.g with reference to the vital role of thirst and hunger!) and in other, the neuroanatomy of inner brain is at the level of specialist and detailed with little description of connectivity and functional anatomy (e.g. the section on habenula). I would suggest that section 4.3 be rewritten covering the same contents but with the consistency and structural narrative of the rest of the paper.
Other minor issues are listed below and are easily fixable. However, in general, quite a few statements in the review are not supported by evidence. some examples are shown below.
Line no. comment
38 full stop needed
44 who and McEwen?
88 in this section, it may be useful to introduce stress in contrast to anxiety and arousal. In other words, describe how anxiety and arousal stand against stress.
92 has since been expanded
103 what do you mean by ‘critical’ brain regions? Are non-critical brain regions?
110 the same issue, what is meant by ‘crucial’ brain regions?
113 citations needed
140 citations needed
153 replace that posits that with and posits that
170 is figure 2 really necessary?
176 see 103 comment
180 define adversity
193 citations needed
210 detected by?
214 what age range?
217 aversion to what?
240 the change between rodents and human is rather confusing. You need to clarify who you are talking about. Reference 47 is about rodents and appears in the middle of discussion about children (human)
258 citations?
260 delete however
345 citations
404 many biases?
443 why executive summary and not summary?
463 NAcc?
471-5 no necessary to explain the importance of water and food and other basic needs
491 citations
557 citations
581 citations also highlight why theta oscillation is important
601 citations also describe Morris water maze learning paradigm
Author Response
March 18, 2025
Marta Pardo, PhD
Guest Editor
International Journal of Molecular Sciences
Dear Dr. Marta Pardo,
Subject: “A Comprehensive Overview of Stress, Resilience and Neuroplasticity Mechanisms”. Manuscript No. ijms- 3506468.
Thank you for your email, which includes the reviewers’ comments. We have carefully reviewed the feedback and revised the manuscript accordingly. Our point-by-point responses are provided below. The changes in the manuscript have been highlighted to indicate all major edits. The corrections and suggestions you provided have significantly improved the paper. We hope that the revised version is now suitable for publication and look forward to your response in due course.
Additionally, we would like to inform you that we have added a new author, Dr. Germán Vega-Flores, to the manuscript. This addition reflects his significant contributions to writing, reviewing, and editing the final, improved version of the manuscript. I have attached the Authorship Change Form, signed by all authors, for your reference.
Sincerely,
Mario Buenrostro-Jauregui, Ph.D.
Universidad Iberoamericana
México City
México
mario.buenrostro@ibero.mx
General statements about the revised manuscript
We have strived to address most of the reviewers' comments. We addressed those comments that we felt were the most relevant and those that were relevant to the focus of our article. In other cases, we justify the reasons for noncompliance in the point-by-point response to comments section of this letter (see below). We also performed minor modifications throughout the article in addition to those kindly recommended by the reviewers to improve clarity and accuracy. All the authors agreed with the changes in the manuscript. All the revisions/modifications of the manuscript were highlighted using the MS Word “Track Changes” tool for easy identification. An academic English editing service was hired to revise and proofread the manuscript. All the changes in the manuscript made by both the editor and us are highlighted. Below, you will find point-by-point responses to your comments. Regular font style is used for featured comments, and italics have been used to show our responses.
Point-by-Point Responses to the Reviewers’ Comments
REVIEWER 2:
In some parts the information is rather basic (e.g with reference to the vital role of thirst and hunger!) and in other, the neuroanatomy of inner brain is at the level of specialist and detailed with little description of connectivity and functional anatomy (e.g. the section on habenula).
RESPONSE: We greatly appreciate the feedback received from the reviewer. We rewrote many parts of the article and added more than 20 additional references. Due to the extensive nature of the changes, we kindly ask the reviewer to check the changes made in the manuscript with the "Track Changes" option activated in Word.
I would suggest that section 4.3 be rewritten covering the same contents but with the consistency and structural narrative of the rest of the paper.
RESPONSE: We rewrote Section 4.3 to make the description more consistent across the entire manuscript and in this section in particular.
Other minor issues are listed below and are easily fixable. However, in general, quite a few statements in the review are not supported by evidence. some examples are shown below.
Line no. comment
38 full stop needed
RESPONSE: This issue was resolved.
44 who and McEwen?
RESPONSE: This issue was resolved.
88 in this section, it may be useful to introduce stress in contrast to anxiety and arousal. In other words, describe how anxiety and arousal stand against stress.
RESPONSE: This issue was resolved. We added citations and modified the text as follows: “Stress, arousal, and anxiety are interrelated but distinct psychological constructs that impact cognition, behavior, and performance. Briefly, stress is a process involving the perception of and response to challenging or threatening stimuli[12,13]. Arousal is an acute physiological component of the stress-related response, activating the autonomic nervous system and releasing hormones to prepare the body for action[14]. In contrast, anxiety can arise as a consequence of prolonged or repeated stress, and is linked to specific emotional states and brain processes[14,15]. Chronic stress patterns induce anxiety-like behaviors. In summary, arousal is an immediate physiological response to stress, whereas anxiety can be an emotional state that follows chronic stress. In particular, the definition of stress has evolved over time in association with advancements in the field.”
92 has since been expanded
RESPONSE: This issue was resolved.
103 what do you mean by ‘critical’ brain regions? Are non-critical brain regions?
RESPONSE: This issue was resolved.
110 the same issue, what is meant by ‘crucial’ brain regions?
RESPONSE: This issue was resolved.
113 citations needed
RESPONSE: This issue was resolved. We added the correct citation.
140 citations needed
RESPONSE: This issue was resolved. We added the correct citation.
153 replace that posits that with and posits that
RESPONSE: This issue was resolved.
170 is figure 2 really necessary?
RESPONSE: Figure 2 is essential for presenting a clearer and more schematic representation of the integrated model of early-life stress and its long-term effects. By integrating the three different hypotheses that have been proposed to explain the outcomes of early-life stress, the figure facilitates a more accessible understanding of how these theoretical perspectives intersect. Additionally, it visually organizes the possible psychological, physiological, and behavioral responses in adulthood, making the complex interactions easier to interpret. This graphical representation enhances comprehension by summarizing key relationships that might be difficult to convey solely through text.
176 see 103 comment
RESPONSE: This issue was resolved.
180 define adversity
RESPONSE: This issue was resolved. We added the following text: “Adversity is a fundamental concept for understanding the topics we will explore later. It refers to difficult and harmful events or conditions that threaten an individual's well-being[31,33]. Adversity includes a wide range of experiences, ranging from the difficulty and unpredictability of the environment to psychosocial stressors such as abuse, neglect, loss, and conflict[41]. Exposure to adversity, particularly in early life, can have lasting effects on development and health, increasing susceptibility to mental and physical health problems[31,41,42].”
193 citations needed
RESPONSE: This issue was resolved. We added citations and modified the text as follows: “Factors such as maternal stress and adversity affect fetal brain development possibly through plasticity mechanisms such as neurogenesis and synaptogenesis[35,39–41]”.
210 detected by?
RESPONSE: This issue was resolved. We modified the text as follows: “For example, institutionalized children show disorganized attachment styles, a reduced brain volume, and functional differences detected by electroencephalography (EEG) [24,57].”
214 what age range?
RESPONSE: In the same paragraph, we mentioned that the study was performed with 262 children aged 8 to –16 years.
217 aversion to what?
RESPONSE: This issue was resolved. We modified the text as follows: “In animal models, a lack of maternal interaction alters fear learning and amygdala activation, whereas maternal presence blocks these stress responses [53].”
240 the change between rodents and human is rather confusing. You need to clarify who you are talking about. Reference 47 is about rodents and appears in the middle of discussion about children (human)
RESPONSE: This issue was resolved. We modified the text in the full paragraph as follows: “Childhood is a critical period of brain development, and even less menacing situations can act as stressors that can shape neural circuitry. In childhood, entering school introduces new sources of stimuli, such as teachers and peers, which can be both enriching and adverse [63]. Children in unpredictable environments exhibit lower effortful control, greater cognitive flexibility, and biases toward negative stimuli[64]. Also, in a study of Sprague–Dawley rats, researchers found that chronic exposure to threats, such as bullying, impacts brain development, leading to increased dendritic spine growth in the amygdala and atrophy in the hippocampus and PFC [65].”
258 citations?
RESPONSE: This issue was resolved. We added citations and modified the text as follows: “Animal and human models have shown that early adversity not only has immediate repercussions but can also transmit negative effects to future generations, as evidenced by epigenetic alterations and changes in brain structure [62,63].”
260 delete however
RESPONSE: This issue was resolved.
345 citations
RESPONSE: This issue was resolved. We added citations and modified the text as follows: “Three main approaches have been used to model vulnerability and resilience: selective breeding, selection of subpopulations, and transgenic models[31]. We describe them below.”
404 many biases?
RESPONSE: This issue was resolved.
443 why executive summary and not summary?
RESPONSE: This issue was resolved.
463 NAcc?
RESPONSE: This issue was resolved.
471-5 no necessary to explain the importance of water and food and other basic needs
RESPONSE: This issue was resolved. We modified the text of the full paragraph as follows: “Motivated behaviors depend on the reward system due to its relevance to the maintenance biological system homeostasis.. Reinforced behaviors ensure both group and individual survival. Some primary motivated behaviors are sex, hunger and thirst that are necessary for the survival of the individual or the species. On the other hand, sex seems to have no relevance to the survival of the individual; however, it is necessary for reproduction in species that use sexual reproduction.”
491 citations
RESPONSE: This issue was resolved. We have added citations and modified the text as follows: “In the consummatory phase, the release of OPIs increases, and that of DAs decreases; specifically, opioids are released from the beginning of the consummatory phase until shortly after the end of the phase[105,108].”
557 citations
RESPONSE: This issue was resolved. We added citations.
581 citations also highlight why theta oscillation is important
RESPONSE: This issue was resolved. We added citations and modified the text as follows: “This study also revealed hyperactivation of theta power when recorded with electrodes implanted in the ventral hippocampus during exposure to novel environments. Theta brain waves are patterns of electrical activity in the brain that oscillate in the frequency range of 4 to 8 Hz. These waves are associated with various cognitive and behavioral functions and mental states such as memory, learning, and spatial navigation[126]. An overactivation of theta waves refers to an abnormal increase in the amplitude or frequency of these oscillations. An excessive or dysregulated overactivation of theta waves is associated with impaired cognitive and emotional functions.[41,125]
601 citations also describe Morris water maze learning paradigm
RESPONSE: This issue was resolved. We added citations and modified the text as follows: “In male mice, the survival of new neurons in adulthood is reduced, which is correlated with poor performance in the object recognition task and Morris water maze. The Morris water maze learning paradigm is a widely used behavioral test in rodent research (primarily in rats and mice) to assess hippocampus-dependent spatial learning and memory. It is based on the animals' ability to learn and remember the location of a hidden platform in a pool filled with opaque water using spatial cues from the environment [129].”
Round 2
Reviewer 1 Report
Comments and Suggestions for Authors
In this version of the reveiw, “A Comprehensive Overview of Stress, Resilience and Neuroplasticity Mechanisms.” We can see an acceptable evolution compared to the first version because it has become more structured with more explanation.
the authors have taken the reviewer's remarks and suggestions into consideration, which has positively impacted the quality and consistency of the review.
with this version, the review shows an excellent scientific level and represents an added value in the interested research topics
the article is accepted for me with this version
Reviewer 2 Report
Comments and Suggestions for Authors
Please check syntax. There are a few minor errors.